# Similarity Component Analysis

**Soravit Changpinyo**[*]
Dept. of Computer Science
U. of Southern California
Los Angeles, CA 90089
schangpi@usc.edu

**Kuan Liu**[*]
Dept. of Computer Science
U. of Southern California
Los Angeles, CA 90089
kuanl@usc.edu

**Fei Sha**
Dept. of Computer Science
U. of Southern California
Los Angeles, CA 90089
feisha@usc.edu

## Abstract

Measuring similarity is crucial to many learning tasks. To this end, metric learning has been the dominant paradigm. However, similarity is a richer and broader notion than what metrics entail. For example, similarity can arise from the process of aggregating the decisions of multiple latent components, where each latent component compares data in its own way by focusing on a different subset of features. In this paper, we propose Similarity Component Analysis (SCA), a probabilistic graphical model that discovers those latent components from data. In SCA, a latent component generates a local similarity value, computed with its own metric, independently of other components. The final similarity measure is then obtained by combining the local similarity values with a (noisy-)OR gate. We derive an EM-based algorithm for fitting the model parameters with similarity-annotated data from pairwise comparisons. We validate the SCA model on synthetic datasets where SCA discovers the ground-truth about the latent components. We also apply SCA to a multiway classification task and a link prediction task. For both tasks, SCA attains significantly better prediction accuracies than competing methods. Moreover, we show how SCA can be instrumental in exploratory analysis of data, where we gain insights about the data by examining patterns hidden in its latent components' local similarity values.

## 1 Introduction

Learning how to measure similarity (or dissimilarity) is a fundamental problem in machine learning. Arguably, if we have the right measure, we would be able to achieve a perfect classification or clustering of data. If we parameterize the desired dissimilarity measure in the form of a metric function, the resulting learning problem is often referred to as *metric learning*. In the last few years, researchers have invented a plethora of such algorithms [18, 5, 11, 13, 17, 9]. Those algorithms have been successfully applied to a wide range of application domains.

However, the notion of (dis)similarity is much richer than what metric is able to capture. Consider the classical example of CENTAUR, MAN and HORSE. MAN is similar to CENTAUR and CENTAUR is similar to HORSE. Metric learning algorithms that model the two similarities well would need to assign small distances among those two pairs. On the other hand, the algorithms will also need to strenuously battle against assigning a small distance between MAN and HORSE due to the triangle inequality, so as to avoid the fallacy that MAN is similar to HORSE too! This example (and others [12]) thus illustrates the important properties, such as non-transitiveness and non-triangular inequality, of (dis)similarity that metric learning has not adequately addressed.

Representing objects as points in high-dimensional feature spaces, most metric learning learning algorithms assume that the same set of features contribute indistinguishably to assessing similarity. In

---

[*]Equal contributions

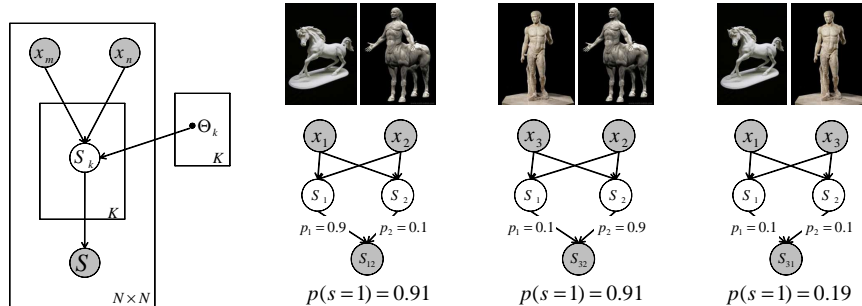

Figure 1: Similarity Component Analysis and its application to the example of CENTAUR, MAN and HORSE. SCA has K latent components which give rise to local similarity values $s_k$ conditioned on a pair of data $\boldsymbol{x}_m$ and $\boldsymbol{x}_n$. The model's output $s$ is a combination of all local values through an OR model (straightforward to extend to a noisy-OR model). $\boldsymbol{\Theta}_k$ is the parameter vector for $p(s_k|\boldsymbol{x}_m, \boldsymbol{x}_n)$. See texts for details.

particular, the popular Mahalanobis metric weights each feature (and their interactions) additively when calculating distances. In contrast, similarity can arise from a complex aggregation of comparing data instances on multiple subsets of features, to which we refer as *latent components*. For instance, there are multiple reasons for us to rate two songs being similar: being written by the same composers, being performed by the same band, or of the same genre. For an arbitrary pair of songs, we can rate the similarity between them based on one of the many components or an arbitrary subset of components, while ignoring the rest. Note that, in the learning setting, we observe only the aggregated results of those comparisons — which components are used is latent.

Multi-component based similarity exists also in other types of data. Consider a social network where the network structure (i.e., links) is a supposition of multiple networks where people are connected for various organizational reasons: school, profession, or hobby. It is thus unrealistic to assume that the links exist due to a single cause. More appropriately, social networks are "multiplex" [6, 15].

In this paper, we propose Similarity Component Analysis (SCA) to model the richer similarity relationships beyond what current metric learning algorithms can offer. SCA is a Bayesian network, illustrated in Fig. 1. The similarity (node s) is modeled as a probabilistic combination of multiple latent components. Each latent component ($s_k$) assigns a local similarity value to whether or not two objects are similar, inferring from only a subset (but unknown) of features. The (local) similarity values of those latent components are aggregated with a (noisy-) OR model. Intuitively, two objects are likely to be similar if they are considered to be similar by at least one component. Two objects are likely to be dissimilar if none of the components voices up.

We derive an EM-based algorithm for fitting the model with data annotated with similarity relationships. The algorithm infers the intermediate similarity values of latent components and identifies the parameters for the (noisy-)OR model, as well as each latent component's conditional distribution, by maximizing the likelihood of the training data.

We validate SCA on several learning tasks. On synthetic data where ground-truth is available, we confirm SCA's ability in discovering latent components and their corresponding subsets of features. On a multiway classification task, we contrast SCA to state-of-the-art metric learning algorithms and demonstrate SCA's superior performance in classifying data samples. Finally, we use SCA to model the network link structures among research articles published at NIPS proceedings. We show that SCA achieves the best link prediction accuracy among competitive algorithms. We also conduct extensive analysis on how learned latent components effectively represent link structures.

In section 2, we describe the SCA model and inference and learning algorithms. We report our empirical findings in section 3. We discuss related work in section 4 and conclude in section 5.

## 2 Approach

We start by describing in detail Similarity Component Analysis (SCA), a Bayesian network for modeling similarity between two objects. We then describe the inference procedure and learning algorithm for fitting the model parameters with similarity-annotated data.

## 2.1 Probabilistic model of similarity

In what follows, let $(\boldsymbol{u}, \boldsymbol{v}, s)$ denote a pair of D-dimensional data points $\boldsymbol{u}, \boldsymbol{v} \in \mathbb{R}^D$ and their associated value of similarity $s \in \{\text{DISSIMILAR}, \text{SIMILAR}\}$ or $\{0, 1\}$ accordingly. We are interested in modeling the process of assigning $s$ to these two data points. To this end, we propose Similarity Component Analysis (SCA) to model the conditional distribution $p(s|\boldsymbol{u}, \boldsymbol{v})$, illustrated in Fig. 1.

In SCA, we assume that $p(s|\boldsymbol{u}, \boldsymbol{v})$ is a mixture of multiple latent components's local similarity values. Each latent component evaluates its similarity value independently, using only a *subset* of the D features. Intuitively, there are multiple reasons of annotating whether or not two data instances are similar and each reason focuses *locally* on one aspect of the data, by restricting itself to examining only a different subset of features.

**Latent components** Formally, let $\boldsymbol{u}_{[k]}$ denote the subset of features from $\boldsymbol{u}$ corresponding to the $k$-th latent component where $[k] \subset \{1, 2, \ldots, D\}$. The similarity assessment $s_k$ of this component *alone* is determined by the distance between $\boldsymbol{u}_{[k]}$ and $\boldsymbol{v}_{[k]}$

$$d_k = (\boldsymbol{u} - \boldsymbol{v})^{\mathrm{T}} \boldsymbol{M}_k (\boldsymbol{u} - \boldsymbol{v}) \tag{1}$$

where $\boldsymbol{M}_k \succeq 0$ is a $D \times D$ positive semidefinite matrix, used to measure the distance more flexibly than the standard Euclidean metric. We restrict $\boldsymbol{M}_k$ to be sparse, in particular, only the corresponding $[k]$-th rows and columns are non-zeroes. Note that in principle $[k]$ needs to be inferred from data, which is generally hard. Nonetheless, we have found that empirically even without explicitly constraining $\boldsymbol{M}_k$, we often obtain a sparse solution.

The distance $d_k$ is transformed to the probability for the Bernoulli variable $s_k$ according to

$$P(s_k = 1 | \boldsymbol{u}, \boldsymbol{v}) = (1 + e^{-b_k})[1 - \sigma(d_k - b_k)] \tag{2}$$

where $\sigma(\cdot)$ is the sigmoid function $\sigma(t) = (1 + e^{-t})^{-1}$ and $b_k$ is a bias term. Intuitively, when the (biased) distance $(d_k - b_k)$ is large, $s_k$ is less probable to be 1 and the two data points are regarded less similar. Note that the constraint $\boldsymbol{M}_k$ being positive semidefinite is important as this will constrain the probability to be bounded above by 1.

**Combining local similarities** Assume that there are K latent components. *How can we combine all the local similarity assessments?* In this work, we use an OR-gate. Namely,

$$P(s = 1 | s_1, s_2, \cdots, s_K) = 1 - \prod_{k=1}^{K} \mathbb{I}[s_k = 0] \tag{3}$$

Thus, the two data points are similar ($s = 1$) if *at least* one of the aspects deems so, corresponding to $s_k = 1$ for a particular $k$. The OR-model can be extended to the noisy-OR model [14]. To this end, we model the non-deterministic effect of each component on the final similarity value,

$$P(s = 1 | s_k = 1) = \tau_k = 1 - \theta_k, \quad P(s = 1 | s_k = 0) = 0 \tag{4}$$

In essence, the uncertainty comes from our probability of failure $\theta_k$ (false negative) to identify the similarity if we are only allowed to consider one component at a time. If we can consider all components at the same time, this failure probability would be reduced. The noisy-OR model captures precisely this notion:

$$P(s = 1 | s_1, s_2, \cdots, s_K) = 1 - \prod_{k=1}^{K} \theta_k^{\mathbb{I}[s_k = 1]} \tag{5}$$

where the more $s_k = 1$, the less the false-negative rate is after combination. Note that the noisy-OR model reduces to the OR-model eq. (3) when $\theta_k = 0$ for all $k$.

**Similarity model** Our desired model for the conditional probability $p(s|\boldsymbol{u}, \boldsymbol{v})$ is obtained by marginalizing all possible configurations of the latent components $\boldsymbol{s} = \{s_1, s_2, \cdots, s_K\}$

$$P(s = 0 | \boldsymbol{u}, \boldsymbol{v}) = \sum_{\boldsymbol{s}} P(s = 0 | \boldsymbol{s}) \prod_k P(s_k | \boldsymbol{u}, \boldsymbol{v}) = \sum_{\boldsymbol{s}} \prod_k \theta_k^{\mathbb{I}[s_k = 1]} P(s_k | \boldsymbol{u}, \boldsymbol{v})$$

$$= \prod_k [\theta_k p_k + 1 - p_k] = \prod_k [1 - \tau_k p_k] \tag{6}$$

where $p_k = p(s_k = 1 | \boldsymbol{u}, \boldsymbol{v})$ is a shorthand for eq. (2). Note that despite the exponential number of configurations for $\boldsymbol{s}$, the marginalized probability is tractable. For the OR-model where $\theta_k = 0$, the conditional probability simplifies to $P(s = 0 | \boldsymbol{u}, \boldsymbol{v}) = \prod_k [1 - p_k]$.

## 2.2 Inference and learning

Given an annotated training dataset $\mathcal{D} = \{(\boldsymbol{x}_m, \boldsymbol{x}_n, s_{mn})\}$, we learn the parameters, which include all the positive semidefinite matrices $\boldsymbol{M}_k$, the biases $b_k$ and the false negative rates $\theta_k$ (if noisy-OR is used), by maximizing the likelihood of $\mathcal{D}$. Note that we will assume that K is known throughout this work. We develop an EM-style algorithm to find the local optimum of the likelihood.

**Posterior** The posteriors over the hidden variables are computationally tractable:

$$
\begin{aligned}
q_k &= P(s_k = 1 | \boldsymbol{u}, \boldsymbol{v}, s = 0) = \frac{p_k \theta_k \prod_{l \neq k} [1 - \tau_l p_l]}{P(s = 0 | \boldsymbol{u}, \boldsymbol{v})} \\
r_k &= P(s_k = 1 | \boldsymbol{u}, \boldsymbol{v}, s = 1) = \frac{p_k \left(1 - \theta_k \prod_{l \neq k} [1 - \tau_l p_l]\right)}{P(s = 1 | \boldsymbol{u}, \boldsymbol{v})}
\end{aligned}
\tag{7}
$$

For OR-model eq. (3), these posteriors can be further simplified as all $\theta_k = 0$.

Note that, these posteriors are sufficient to learn the parameters $\boldsymbol{M}_k$ and $b_k$. To learn the parameters $\theta_k$, however, we need to compute the expected likelihood with respect to the posterior $P(\boldsymbol{s} | \boldsymbol{u}, \boldsymbol{v}, s)$. While this posterior is tractable, the expectation of the likelihood is not and variational inference is needed [10]. We omit the derivation for brevity. In what follows, we focus on learning $\boldsymbol{M}_k$ and $b_k$.

For the $k$-th component, the relevant terms in the expected log-likelihood, given the posteriors, from a single similarity assessment $s$ on $(\boldsymbol{u}, \boldsymbol{v})$, is

$$
J_k = q_k^{1-s} r_k^s \log P(s_k = 1 | \boldsymbol{u}, \boldsymbol{v}) + (1 - q_k^{1-s} r_k^s) \log(1 - P(s_k = 1 | \boldsymbol{u}, \boldsymbol{v}))
\tag{8}
$$

**Learning the parameters** Note that $J_k$ is not jointly convex in $b_k$ and $\boldsymbol{M}_k$. Thus, we optimize them alternatively. Concretely, fixing $\boldsymbol{M}_k$, we grid search and optimize over $b_k$. Fixing $b_k$, maximizing $J_k$ with respect to $\boldsymbol{M}_k$ is convex optimization as $J_k$ is a concave function in $\boldsymbol{M}_k$ given the linear dependency of the distance eq. (1) on this parameter.

We use the method of projected gradient ascent. Essentially, we take a gradient ascent step to update $\boldsymbol{M}_k$ iteratively. If the update violates the positive semidefinite constraint, we project back to the feasible region by setting all negative eigenvalues of $\boldsymbol{M}_k$ to zeroes. Alternatively, we have found that reparameterizing $J_k$ in the following form $\boldsymbol{M}_k = \boldsymbol{L}_k^{\mathsf{T}} \boldsymbol{L}_k$ is more computationally advantageous, as $\boldsymbol{L}_k$ is unconstrained. We use L-BFGS to optimize with respect to $\boldsymbol{L}_k$ and obtain faster convergence and better objective function values. (While this procedure only guarantees local optima, we observe no significant detrimental effect of arriving at those solutions.) We give the exact form of gradients with respect to $\boldsymbol{M}_k$ and $\boldsymbol{L}_k$ in the Suppl. Material.

## 2.3 Extensions

**Variants to local similarity models** The choice of using logistic-like functions eq. (2) for modeling local similarity of the latent components is orthogonal to how those similarities are combined in eq. (3) or eq. (5). Thus, it is relatively straightforward to replace eq. (2) with a more suitable one. For instance, in some of our empirical studies, we have constrained $\boldsymbol{M}_k$ to be a diagonal matrix with nonnegative diagonal elements. This is especially useful when the feature dimensionality is extremely high. We view this flexibility as a modeling advantage.

**Disjoint components** We could also explicitly express our desiderata that latent components focus on non-overlapping features. To this end, we penalize the likelihood of the data with the following regularizer to promote disjoint components

$$
R(\{\boldsymbol{M}_k\}) = \sum_{k,k'} \mathsf{diag}(\boldsymbol{M}_k)^{\mathsf{T}} \mathsf{diag}(\boldsymbol{M}_{k'})
\tag{9}
$$

where $\mathsf{diag}(\cdot)$ extracts the diagonal elements of the matrix. As the metrics are constrained to be positive semidefinite, the inner product attains its minimum of zero when the diagonal elements, which are nonnegative, are orthogonal to each other. This will introduce zero elements on the diagonals of the metrics, which will in turn deselect the corresponding feature dimensions, because the corresponding rows and columns of those elements are necessarily zero due to the positive semidefinite constraints. Thus, metrics that have orthogonal diagonal vectors will use non-overlapping subsets of features.

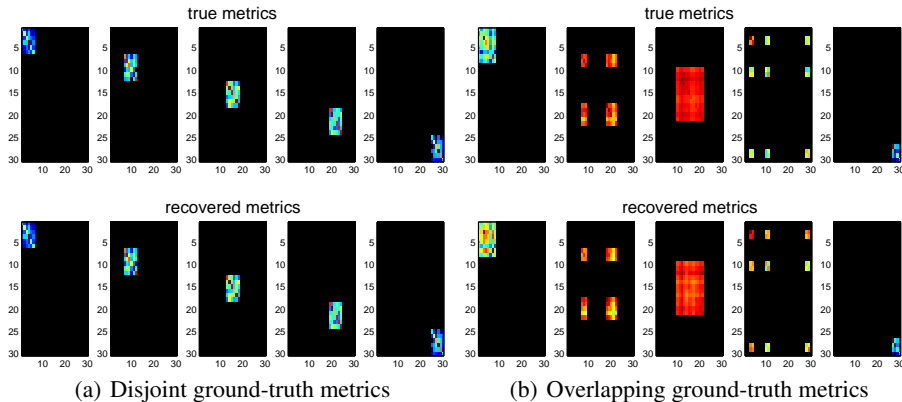

<div align="center">(a) Disjoint ground-truth metrics      (b) Overlapping ground-truth metrics</div>

Figure 2: On synthetic datasets, SCA successfully identifies the sparse structures and (non)overlapping patterns of ground-truth metrics. See texts for details. Best viewed in color.

## 3 Experimental results

We validate the effectiveness of SCA in modeling similarity relationships on three tasks. In section 3.1, we apply SCA to synthetic datasets where the ground-truth is available to confirm SCA's ability in identifying correctly underlying parameters. In section 3.2, we apply SCA to a multiway classification task to recognize images of handwritten digits where similarity is equated to having the same class label. SCA attains superior classification accuracy to state-of-the-art metric learning algorithms. In section 3.3, we apply SCA to a link prediction problem for a network of scientific articles. On this task, SCA outperforms competing methods significantly, too.

Our baseline algorithms for modeling similarity are information-theoretic metric learning (ITML) [5] and large margin nearest neighbor (LMNN) [18]. Both methods are discriminative approaches where a metric is optimized to reduce the distances between data points from the same label class (or similar data instances) and increase the distances between data points from different classes (or dissimilar data instances). When possible, we also contrast to multiple metric LMNN (MM-LMNN) [18], a variant to LMNN where multiple metrics are learned from data.

### 3.1 Synthetic data

**Data** We generate a synthetic dataset according to the graphical model in Fig. 1. Specifically, our feature dimensionality is $D = 30$ and the number of latent components is $K = 5$. For each component $k$, the corresponding metric $\boldsymbol{M}_k$ is a $D \times D$ sparse positive semidefinite matrix where only elements in a $6 \times 6$ matrix block on the diagonal are nonzero. Moreover, for different $k$, these block matrices do not overlap in rows and columns indices. In short, these metrics mimic the setup where each component focuses on its own $1/K$-th of total features that are disjoint from each other. The first row of Fig. 2(a) illustrates these 5 matrices while the black background color indicates zero elements. The values of nonzero elements are randomly generated as long as they maintain the positive semidefiniteness of the metrics. We set the bias term $b_k$ to zeroes for all components. We sample $N = 500$ data points randomly from $\mathbb{R}^D$. We select a random pair and compute their similarity according to eq. (6) and threshold at 0.5 to yield a binary label $s \in \{0, 1\}$. We select randomly 74850 pairs for training, 24950 for development, 24950 for testing.

**Method** We use the OR-model eq. (3) to combine latent components. We evaluate the results of SCA on two aspects: how well we can recover the ground-truth metrics (and biases) and how well we can use the parameters to predict similarities on the test set.

**Results** The second row of Fig. 2(a) contrasts the learned metrics to the ground-truth (the first row). Clearly, these two sets of metrics have almost identical shapes and sparse structures. Note that for this experiment, we did not use the disjoint regularizer (described in section 2.3) to promote sparsity and disjointness in the learned metrics. Yet, the SCA model is still able to identify those structures. For the biases, SCA identifies them as being close to zero (details are omitted for brevity).

Table 1: Similarity prediction accuracies and standard errors (%) on the synthetic dataset

| BASELINES | | SCA | | | | | |
|---|---|---|---|---|---|---|---|
| ITML | LMNN | K = 1 | K = 3 | K = 5 | K = 7 | K = 10 | K = 20 |
| 72.7±0.0 | 71.3±0.2 | 72.8±0.0 | 82.1±0.1 | **91.5±0.1** | **91.7±0.1** | **91.8±0.1** | 90.2±0.4 |

Table 2: Misclassification rates (%) on the MNIST recognition task

| D | BASELINES | | | | SCA | | |
|---|---|---|---|---|---|---|---|
| | EUC. | ITML | LMNN | MM-LMNN | K = 1 | K = 5 | K = 10 |
| 25 | 21.6 | 15.1 | 20.6 | 20.2 | $17.7 \pm 0.9$ | $16.0 \pm 1.5$ | $\mathbf{14.5 \pm 0.6}$ |
| 50 | 18.7 | 13.35 | 16.5 | 13.6 | $13.8 \pm 0.3$ | $12.0 \pm 1.1$ | $\mathbf{11.4 \pm 0.6}$ |
| 100 | 18.1 | 11.85 | 13.4 | **9.9** | $12.1 \pm 0.1$ | $10.8 \pm 0.6$ | $11.1 \pm 0.3$ |

Table 1 contrasts the prediction accuracies by SCA to competing methods. Note that ITML, LMNN and SCA with $K = 1$ perform similarly. However, when the number of latent components increases, SCA outperforms other approaches by a large margin. Also note that when the number of latent components exceeds the ground-truth $K = 5$, SCA reaches a plateau until overfitting.

In real-world data, "true metrics" may overlap, that is, it is possible that different components of similarity rely on overlapping set of features. To examine SCA's effectiveness in this scenario, we create another synthetic data where true metrics heavily overlap, illustrated in the first row of Fig. 2(b). Nonetheless, SCA is able to identify the metrics correctly, as seen in the second row.

## 3.2 Multiway classification

For this task, we use the MNIST dataset, which consists of 10 classes of hand-written digit images. We use PCA to reduce the original dimension from 784 to $D = 25, 50$ and $100$, respectively. We use 4200 examples for training, 1800 for development and 2000 for testing.

The data is in the format of $(\boldsymbol{x}_n, y_n)$ where $y_n$ is the class label. We convert them into the format $(\boldsymbol{x}_m, \boldsymbol{x}_n, s_{mn})$ that SCA expects. Specifically, for every *training* data point, we select its 15 nearest neighbors among samples in the same class and formulate 15 similar relationships. For dissimilar relationships, we select its 80 nearest neighbors among samples from the rest classes. For testing, the label $y$ of $\boldsymbol{x}$ is determined by

$$y = \arg\max_c s_c = \arg\max_c \sum_{\boldsymbol{x}' \in B_c(\boldsymbol{x})} P(s = 1 | \boldsymbol{x}, \boldsymbol{x}') \tag{10}$$

where $s_c$ is the similarity score to the $c$-th class, computed as the sum of 5 largest similarity values $B_c$ to samples in that class. In Table 2, we show classification error rates for different values of D. For $K > 1$, SCA clearly outperforms single-metric based baselines. In addition, SCA performs well compared to MM-LMNN, achieving far better accuracy for small D.

## 3.3 Link prediction

We evaluate SCA on the task of link prediction in a "social" network of scientific articles. We aim to demonstrate SCA's power to model similarity/dissimilarity in "multiplex" real-world network data. In particular, we are interested in not only link prediction accuracies, but also the insights about data that we gain from analyzing the identified latent components.

**Setup** We use the NIPS 0-12 dataset [1] to construct the aforementioned network. The dataset contains papers from the NIPS conferences between 1987 and 1999. The papers are organized into 9 sections (topics) (cf. Suppl. Material). We sample randomly 80 papers per section and use them to construct the network. Each paper is a vertex and two papers are connected with an edge and deemed as similar if both of them belong to the same section.

We experiment three representations for the papers: (1) Bag-of-words (BoW) uses normalized occurrences (frequencies) of words in the documents. As a preprocessing step, we remove "rare" words that appear less than 75 times and appear more than 240. Those words are either too specialized (thus generalize poorly) or just functional words. After the removal, we obtain 1067 words. (2) Topic (ToP) uses the documents' topic vectors (mixture weights of topics) after fitting the corpus

Table 3: Link prediction accuracies and their standard errors (%) on a network of scientific papers

| Feature type | BASELINES | | | SCA-DIAG | | SCA | |
|---|---|---|---|---|---|---|---|
| | SVM | ITML | LMNN | K = 1 | K* | K = 1 | K* |
| BoW | 73.3±0.0 | - | - | 64.8 ± 0.1 | **87.0 ± 1.2** | - | - |
| ToW | 75.3±0.0 | - | - | 67.0 ± 0.0 | **88.1 ± 1.4** | - | - |
| ToP | 71.2±0.0 | 81.1±0.1 | 80.7±0.1 | 62.6 ± 0.0 | 81.0 ± 0.8 | 81.0 ± 0.0 | **87.6 ± 1.0** |

to a 50-topic LDA [4]. (3) Topic-words (ToW) is essentially BoW except that we retain only 1036 frequent words used by the topics of the LDA model (top 40 words per topic).

**Methods**  We compare the proposed SCA extensively to several competing methods for link prediction. For BoW and ToW represented data, we compare SCA with diagonal metrics (SCA-DIAG, cf. section 2.3) to Support Vector Machines (SVM) and logistic regression (LOGIT) to avoid high computational costs associated with learning high-dimensional matrices (the feature dimensionality $D \approx 1000$). To apply SVM/LOGIT, we treat the link prediction as a binary classification problem where the input is the absolute difference in feature values between the two data points.

For 50-dimensional ToP represented data, we compare SCA (SCA) and SCA-DIAG to SVM/LOGIT, information-theoretical metric learning (ITML), and large margin nearest neighbor (LMNN).

Note that while LMNN was originally designed for nearest-neighbor based classification, it can be adapted to use similarity information to learn a global metric to compute the distance between any pair of data points. We learn such a metric and threshold on the distance to render a decision on whether two data points are similar or not (i.e., whether there is a link between them). On the other end, multiple-metric LMNN, while often having better classification performance, cannot be used for similarity and link prediction as it does not provide a principled way of computing distances between two arbitrary data points when there are multiple (local) metrics.

**Link or not?**  In Table 3, we report link prediction accuracies, which are averaged over several runs of randomly generated 70/30 splits of the data. SVM and LOGIT perform nearly identically so we report only SVM. For both SCA and SCA-DIAG, we report results when a single component is used as well as when the optimal number of components are used (under columns K*).

Both SCA-DIAG and SCA outperform the rest methods by a significant margin, especially when the number of latent components is greater than 1 (K* ranges from 3 to 13, depending on the methods and the feature types). The only exception is SCA-DIAG with one component (K = 1), which is an overly restrictive model as the diagonal metrics constrain features to be combined additively. This restriction is overcome by using a larger number of components.

**Edge component analysis**  *Why does learning latent components in SCA achieve superior link prediction accuracies*? The (noisy-)OR model used by SCA is naturally inclined to favoring "positive" opinions — a pair of samples are regarded as being similar as long as there is one latent component strongly believing so. This implies that a latent component can be tuned to a specific group of samples if those samples rely on common feature characteristics to be similar.

Fig. 3(a) confirms our intuition. The plot displays in relative strength —darker being stronger — how much each latent component believes a pair of articles from the same section should be similar. Concretely, after fitting a 9-component SCA (from documents in ToP features), we consider edges connecting articles in the same section and compute the average local similarity values assigned by each component. We observe two interesting sparse patterns: for each section, there is a dominant latent component that strongly supports the fact that the articles from that section should be similar (e.g., for section 1, the dominant one is the 9-th component). Moreover, for each latent component, it often strongly "voices up" for one section – the exception is the second component which seems to support both section 3 and 4. Nonetheless, the general picture is that, each section has a signature in terms of how similarity values are distributed across latent components.

This notion is further illustrated, with greater details, in Fig. 3(b). While Fig. 3(a) depicts averaged signature for each section, the scatterplot displays $2D$ embeddings computed with the t-SNE algorithm, on each individual edge's signature — 9-dimensional similarity values inferred with the 9 latent components. The embeddings are very well organized in 9 clusters, colored with section IDs.

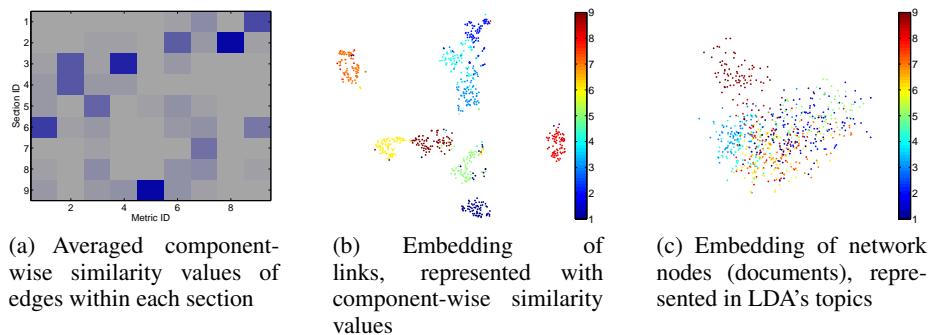

(a) Averaged component-wise similarity values of edges within each section

(b) Embedding of links, represented with component-wise similarity values

(c) Embedding of network nodes (documents), represented in LDA's topics

Figure 3: Edge component analysis. Representing network links with local similarity values reveals interesting structures, such as nearly one-to-one correspondence between latent components and sections, as well as clusters. However, representing articles in LDA's topics does not reveal useful clustering structures such that links can be inferred. See texts for details. Best viewed in color.

In contrast, embedding documents using their topic representations does not reveal clear clustering structures such that network links can be inferred. This is shown in Fig. 3(c) where each dot corresponds to a document and the low-dimensional coordinates are computed using t-SNE (symmetrized KL divergence between topics is used as a distance measure). We observe that while topics themselves do not reveal intrinsic (network) structures, latent components are able to achieve so by applying highly-specialized metrics to measure local similarities and yield characteristic signatures.

We also study whether or not the lack of an edge between a pair of dissimilar documents from different sections, can give rise to characteristic signatures from the latent components. In summary, we do not observe those telltale signatures for those pairs. Detailed results are in the Suppl. Material.

## 4  Related Work

Our model learns multiple metrics, one for each latent component. However, the similarity (or associated dissimilarity) from our model is definitely non-metric due to the complex combination. This stands in stark contrast to most metric learning algorithms [19, 8, 7, 18, 5, 11, 13, 17, 9].

[12] gives an information-theoretic definition of (non-metric) similarity as long as there is a probabilistic model for the data. Our approach of SCA focuses on the relationship between data but not data themselves. [16] proposes visualization techniques for non-metric similarity data.

Our work is reminiscent of probabilistic modeling of overlapping communities in social networks, such as the mixed membership stochastic blockmodels [3]. The key difference is that those works model vertices with a mixture of latent components (communities) where we model the interactions between vertices with a mixture of latent components. [2] studies a social network whose edge set is the union of multiple edge sets in hidden similarity spaces. Our work explicitly models the probabilistic process of combining latent components with a (noisy-)OR gate.

## 5  Conclusion

We propose Similarity Component Analysis (SCA) for probabilistic modeling of similarity relationship for pairwise data instances. The key ingredient of SCA is to model similarity as a complex combination of multiple latent components, each giving rise to a local similarity value. SCA attains significantly better accuracies than existing methods on both classification and link prediction tasks.

**Acknowledgements** We thank reviewers for extensive discussion and references on the topics of similarity and learning similarity. We plan to include them as well as other suggested experimentations in a longer version of this paper. This research is supported by a USC Annenberg Graduate Fellowship (S.C.) and the IARPA via DoD/ARL contract # W911NF-12-C-0012. The U.S. Government is authorized to reproduce and distribute reprints for Governmental purposes notwithstanding any copyright annotation thereon. The views and conclusions contained herein are those of the authors and should not be interpreted as necessarily representing the official policies or endorsements, either expressed or implied, of IARPA, DoD/ARL, or the U.S. Government.

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
