[Supplementary Material]

# Similarity Component Analysis: Supplementary Material

**Soravit Changpinyo**[*]
Dept. of Computer Science
U. of Southern California
Los Angeles, CA 90089
schangpi@usc.edu

**Kuan Liu**[*]
Dept. of Computer Science
U. of Southern California
Los Angeles, CA 90089
kuanl@usc.edu

**Fei Sha**
Dept. of Computer Science
U. of Southern California
Los Angeles, CA 90089
feisha@usc.edu

## S1  Details on Model Inference

### S1.1  E-Step: Objective function

In Section 2, we have given the posterior distribution of the (hidden) local similarity value variables. Below, we derive the form of expected complete data log-likelihood conditioned on a pair $\boldsymbol{u}, \boldsymbol{v}$. We use the same notation as in Section 2.

$$
\begin{aligned}
&E_{p(\boldsymbol{s})}[\log(P(s, s_1, \ldots, s_{\mathsf{K}}|\boldsymbol{u}, \boldsymbol{v}))] \\
&= E_{p(\boldsymbol{s})}[\sum_{k=1}^{\mathsf{K}}(\log(P(s_k|\boldsymbol{u}, \boldsymbol{v}))) + \log(P(s|s_1, s_2, ..., s_{\mathsf{K}}))] \\
&= E_{p(\boldsymbol{s})}[\sum_{k=1}^{\mathsf{K}}(s_k \log p_k + (1-s_k)\log(1-p_k)) + s\log(1 - \prod_{k=1}^{\mathsf{K}} \theta_k^{1(s_k=1)}) + (1-s)\sum_{k=1}^{\mathsf{K}} s_k \log \theta_k] \\
&= \sum_{k=1}^{\mathsf{K}}(q_k^{1-s} r_k^s \log p_k + (1 - q_k^{1-s} r_k^s)\log(1-p_k)) \\
&\quad + E_{p(\boldsymbol{s})}[s\log(1 - \prod_{k=1}^{\mathsf{K}} \theta_k^{1(s_k=1)})] + (1-s)\sum_{k=1}^{\mathsf{K}} q_k^{1-s} r_k^s \log \theta_k
\end{aligned}
\tag{S1}
$$

where $p(\boldsymbol{s}) = \Pr(s_1, \ldots, s_{\mathsf{K}}|\boldsymbol{u}, \boldsymbol{v}, s) = \prod_{k=1}^{\mathsf{K}} \Pr(s_k|\boldsymbol{u}, \boldsymbol{v}, s)$. $p_k = P(s_k = 1|\boldsymbol{u}, \boldsymbol{v})$ is given by eq. (2). $q_k = P(s_k = 1|\boldsymbol{u}, \boldsymbol{v}, s = 0)$ and $r_k = P(s_k = 1|\boldsymbol{u}, \boldsymbol{v}, s = 1)$ are given by eq. (7). Note that the last equation uses the fact that $E_{p(\boldsymbol{s})}(s_k) = q_k^{1-s} r_k^s$.

The third term $E_{p(\boldsymbol{s})}[s\log(1 - \prod_{k=1}^{\mathsf{K}} \theta_k^{1(s_k=1)})]$ is not tractable; variational methods can be used, as described in [1].

### S1.2  M-step: Optimization

We give the form of gradients with respect to $\boldsymbol{M}$ and $\boldsymbol{L}$ in the (noisy-)OR model. As given in Section 2 and above, for the $k$-th aspect, the relevant terms in the expected log-likelihood given the posteriors, from a single similarity assessment, are

$$
J_k = w_k \log p_k + (1 - w_k)\log(1 - p_k)
\tag{S2}
$$

where $p_k = P(s_k = 1|\boldsymbol{u}, \boldsymbol{v}) = (1 + e^{-b_k})[1 - \sigma(d_k - b_k)]$ and $w_k$ denotes $q_k^{1-s} r_k^s$.

---

[*]Equal contributions

(a) ToP            (b) ToW

Figure S1: The normalized diagonal values of metrics for $\mathsf{K} = 9$

Taking the derivative with respect to $d_k$ gives us

$$\frac{\partial J_k}{\partial d_k} = -w_k\sigma + (1-w_k)\frac{\sigma(1-\sigma)}{\sigma - (c_k-1)/c_k} \tag{S3}$$

where $\sigma$ is a short form of $\sigma(d_k - b_k)$ and $c_k = 1 + e^{-b_k}$. For two different parameterizations $d_k = (\boldsymbol{u} - \boldsymbol{v})^{\mathrm{T}}\boldsymbol{M}_k(\boldsymbol{u} - \boldsymbol{v})$ and $d_k = (\boldsymbol{u} - \boldsymbol{v})^{\mathrm{T}}\boldsymbol{L}_k^T\boldsymbol{L}_k(\boldsymbol{u} - \boldsymbol{v})$, we have

$$\frac{\partial d_k}{\partial \boldsymbol{M_k}} = (\boldsymbol{u} - \boldsymbol{v})(\boldsymbol{u} - \boldsymbol{v})^T$$

$$\frac{\partial d_k}{\partial \boldsymbol{L_k}} = 2\boldsymbol{L}_k(\boldsymbol{u} - \boldsymbol{v})(\boldsymbol{u} - \boldsymbol{v})^T \tag{S4}$$

It follows that the gradients with respect to $\boldsymbol{M}_k$ and $\boldsymbol{L}_k$ are

$$\frac{\partial J_k}{\partial \boldsymbol{M}_k} = [-w_k\sigma + (1-w_k)\frac{\sigma(1-\sigma)}{\sigma - (c_k-1)/c_k}](\boldsymbol{u} - \boldsymbol{v})(\boldsymbol{u} - \boldsymbol{v})^T$$

$$\frac{\partial J_k}{\partial \boldsymbol{L}_k} = 2[-w_k\sigma + (1-w_k)\frac{\sigma(1-\sigma)}{\sigma - (c_k-1)/c_k}]\boldsymbol{L}_k(\boldsymbol{u} - \boldsymbol{v})(\boldsymbol{u} - \boldsymbol{v})^T \tag{S5}$$

## S2    Additional Information and Results on Link Prediction Tasks

**NIPS 0-12 dataset**    The papers from the NIPS 0-12 dataset (Section 3.3) are organized into 9 sections, shown in Table S1.

**Subset of features picked by sparse metrics**    In Fig. S1, we show the diagonal entries of the metrics in the case of ToP and ToW features, both for $\mathsf{K} = 9$. We can think of the diagonal entries as the *weights* put on such features. Features picked by these metrics seem to be sparse and disjoint – signified by the nonoverlapping spiky structure. This validates our assumption that each latent component evaluates its similarity value using a different *subset* of features. We show corresponding features according to these diagonal weights in Table S2 and Table S3. We observe that, for each metric, its top features (words or topics) tend to appear rarely in the sections that the metric can predict similarity well. In other words, two documents are similar in one aspect often because they both lack a certain kind of features. One explanation is that each component tries to reduce sensitivity when predicting similarity by picking features that are less varied in values.

**Additional results on link prediction accuracies**    Besides the dataset described in the main paper, we report link prediction accuracies on another network (we call **NIPS-3**), also sampled from NIPS 0-12, to further validate the effectiveness of SCA. This network uses only 450 papers from NIPS 1997 to NIPS 1999 and has 16059 edges. In this dataset, the numbers of papers are not balanced across sections — there are significantly more papers in Learning Theory and Algorithms &

| ID | Names | ID | Names | ID | Names |
|---|---|---|---|---|---|
| 1 | Cognitive Science | 4 | Algo. and Arch. | 7 | Visual Processing |
| 2 | Neuroscience | 5 | Implementation | 8 | Applications |
| 3 | Learning Theory | 6 | Speech & Sign. Proc. | 9 | Ctrl., Navi., & Plan. |

Table S1: Section names for NIPS 1987 – 1999

| Metric | Top 5 Features |
|---|---|
| 1 | radio financial trains costs achieve |
| 2 | curve representations image attractor kalman |
| 3 | trained speech signals statistics class |
| 4 | retina robot gate regression size |
| 5 | learning model data state activity |
| 6 | stress margin evidence actor barto |
| 7 | dot views conclusion moody perturbation |
| 8 | implementation hmm kalman rbf vlsi |
| 9 | speech chip period vision regression |

Table S2: Top five ToW features for each metric

Architectures than Implementations and Speech & Signal Processing. Nonetheless, in Table S4, we see again that SCA achieves a significant improvement on the link prediction accuracies, similar to that in Table 3 in the main text.

**Additional results on edge component analysis** In Section 3.3, we have observed characteristic signatures from the latent components that result from edges between similar documents. One natural question to ask is whether or not the lack of edges between dissimilar documents (in this case, those from different sections) can give rise to such signatures, too. In Fig. S2, we show the average component-wise dissimilarity values of edges between different sections (how much each latent component believes a pair of articles from different sections should be dissimilar) for ToW feature type with $K = 9$. Not surprisingly, we do not observe those telltale signatures - for those pairs of data, almost all latent components vote them as being dissimilar strongly. This suggests that those latent components have both high sensitivity and specificity.

# References

[1] Tommi S. Jaakkola and Michael I. Jordan. Variational Probabilistic Inference and the QMR-DT Network. *Journal of Artificial Intelligence Research*, 10(1):291–322, May 1999.

Figure S2: Average component-wise dissimilarity values of edges between different sections. Darker indicates higher dissimilarity values.

| Metric | Top 3 Features |
|---|---|
| 1 | 1: human similarity subjects generalization performance similar |
| | 2: image images object recognition face feature features objects |
| | 3: model motor position control eye movement forward trajectory |
| 2 | 29: motion direction figure velocity optical flow retina time |
| | 18: action state reinforcement policy actions learning reward |
| | 35: visual target system attention location information search |
| 3 | 36: time eeg activity attractor data response brain signal single |
| | 1: human similarity subjects generalization performance similar |
| | 50: representation sequence representations information level |
| 4 | 27: words user context word text information documents query |
| | 29: motion direction figure velocity optical flow retina time |
| | 2: image images object recognition face feature features objects |
| 5 | 16: spike firing information rate time spikes neuron model input |
| | 2: image images object recognition face feature features objects |
| | 23: phase figure adaptation contour segment oscillators segments |
| 6 | 31: kernel vector set support function data regression training |
| | 49: time series prediction signal filter neural gamma kalman |
| | 13: block blocks data time algorithm search program parallel |
| 7 | 8: function functions bound theorem bounds loss error proof |
| | 24: learning error noise training weight generalization teacher |
| | 7: energy correlation binary function correlations population |
| 8 | 13: block blocks data time algorithm search program parallel |
| | 41: recognition character characters digit neural segmentation |
| | 25: language connectionist symbol symbols set rules languages |
| 9 | 22: time call path rl channel problem traffic routing rate paths |
| | 29: motion direction figure velocity optical flow retina time |
| | 30: distribution probability variables approximation distributions |

Table S3: Top three ToP features for each metric

| Feature | BASELINES | | | SCA-DIAG | | SCA | |
|---|---|---|---|---|---|---|---|
| type | SVM | ITML | LMNN | $K = 1$ | $K^*$ | $K = 1$ | $K^*$ |
| BoW | 73.3±0.0 | - | - | $70.5 \pm 0.1$ | $\mathbf{89.5 \pm 0.8}$ | - | - |
| ToW | 80.4±0.0 | - | - | $76.5 \pm 0.1$ | $\mathbf{87.8 \pm 1.1}$ | - | - |
| ToP | 75.1±0.0 | 83.8±0.2 | 75.4±0.3 | $70.5 \pm 0.1$ | $84.3 \pm 0.9$ | $83.6 \pm 0.1$ | $\mathbf{89.6 \pm 1.1}$ |

Table S4: Link prediction accuracies and their standard errors (%) on **NIPS3**