[Reviews · NeurIPS 2013]

Submitted by Assigned_Reviewer_6

The authors propose a model of similarity in which a binary similarity score between two vectors (u,v) is modeled as a noisy-or over a set of (learned) Mahalanobis distances. The motivation being that abstract "similarity" is often non-transitive, and is not well-modeled by a single, fixed distance metric over a vector space. Generally speaking, the paper is clearly written and the method makes intuitive sense. The algorithm seems to work in practice, although the experimental results could be expanded a bit to better illustrate the method. Overall, I enjoyed reading this paper.

A few comments:

- In the introduction, the authors stress the need for supporting multi-faceted models of similarity, but the wording of this section seems a bit careless. The centaur example (line 42) illustrates that "similarity" is non-transitive, but the notion of transitivity does not directly carry over to distances, as some discretization must be applied in order to make a hard similarity assessment from distances. It is easy to construct an example in which a distance metric produces a non-transitive nearest-neighbor connectivity graph. The authors should be more careful in their statement of what exactly is to be gained by using multiple metrics.

- A similar method to model multi-faceted similarity was recently proposed by [1]. Although the context there was visualization (not similarity prediction), the authors should probably cite this work as well.

- Line 125--126: Seems like a typo here. The [k] rows/columns should be the only ones allowed to be NON-zero. It is also unclear whether the sparsity constraint is enforced in this work, or proposed as a future extension.

- Section 3.1: in generating the synthetic dataset, there does not appear to be any processing to ensure consistency in the training set similarities. As a result, the comparison to LMNN may not be entirely fair, as it requires transitive similarity structure (labels).

- Section 3.1: How is a similarity score predicted with LMNN? Is a distance threshold estimated from the training set?

- Table 1: should this table also include accuracy on the second synthetic dataset (figure 2b)?

- Figure 2b: the visualization of the recovered metrics is nice, but it might be good to include a quantitative measure as well, such as the angle/alignment between true and recovered metrics.

- Section 3.2: in the MNIST example, is there much diversity in the recovered metrics (eg, as measured by alignment)? What does the theta vector look like: are the multiple metrics actually used?

- How does SCA-Diag perform on the full (uncompressed) mnist data?

- Table 3: there are no results for itml/lmnn/sca on the BoW/ToW representations. Is this because the data was too high-dimensional? Why not compress by PCA as was done in the MNIST experiment? For completeness, it would be nice to see full metric learning (after PCA) as well as diagonal metric learning (without PCA) on both datasets.
Summary: This paper proposes a probabilistic model of pairwise similarity as noisy-or of Mahalanobis distances between two data points. The paper is interesting, well-motivated and clearly written. The experiments could be a bit more thorough.

Submitted by Assigned_Reviewer_7

This paper proposes a graphical model for measuring similarity, based on the assumption that two objects are similar because some (instead of all) of their components are similar. Following this design, they show empirically their model performs better than recent metric learning approaches, when the number of components (K) is larger than 1.

Overall, I like this paper. The model design captures one of the fundamental issues of similarity measures and differentiate it from the metric learning framework. Despite the fact that it's a graphical model, the approach still cleverly leverages some aspects of the metric learning models. Specifically, the component similarity is essentially a Mahalanobis distance (before using a sigmoid function to make it probability). This design makes sure that the model would generalize the metric learning approaches, which reflects in the experimental results as well (when K=1). For some applications/tasks, when K increases, the performance does improve, which demonstrates that this model fits those cases better.

On the other hand, since the model can be viewed as an extension of the Mahalanobis learning, the scalability of the learning process still seems restricted. For instance, PCA needs to be employed to reduce the dimensionality of the raw data before applying this algorithm (same applied to ITML and LMNN though). Special constraints on M_k need to be enforced (SCA-DIAG, where the component similarity becomes a weighted Euclidean distance [1]) to reduce the complexity. Such issues are not critical, but it would be nice if the authors can further discuss them in the paper. Another issue that I wish the authors could discuss further is the sparseness of M_k. It is mentioned in the paper that M_k is sparse naturally when learning the model. Why this happens is nevertheless not very clear since there seems no special regularization on M_k. It would be interesting if the authors could provide some insight.

Other comments:

1. The example of CENTAUR, MAN and HORSE is in fact the canonical example to show that similarity does not need to enforce triangular inequality, although the authors did not make this argument explicit. See Figure 3 in [2].
2. The component similarity design is especially suitable for word similarity, where each word can have multiple senses. For instance, JAGUAR and AUTO are similar. JAGUAR and PUMA are also similar. However, these two pairs are similar because different components (e.g., senses) of the word JAGUAR. One of the recent papers that specifically addresses this issue is [3]. It would be interesting to know if the proposed model can subsume their approach given.
3. Perhaps it's out of the scope of this paper, but the model design bears some resemblance to the Siamese neural network approach [4,5,6]. It should be very straightforward to design a NN architecture with the component similarity subnetworks, and thus interesting to see how it compares to the graphical model approach.

[1] Schultz and Joachims. Learning a distance metric from relative comparisons. NIPS-2004.
[2] D Lin. An information-theoretic definition of similarity. ICML-1998.
[3] Reisinger and Mooney. Multi-Prototype Vector-Space Models of Word Meaning. NAACL-2010.
[4] Bromley, Bentz, Bottou, Guyon, LeCun, Moore, Sackinger and Shah. Signature verification using a “Siamese” time delay neural network. 1993. International Journal Pattern Recognition and Artificial Intelligence, 7(4):669–688.
[5] Yih, Toutanova, Platt and Meek. Learning Discriminative Projections for Text Similarity Measures. CoNLL-2011.
[6] Weston, Bengio, and Usunier. Large scale image annotation: Learning to rank with joint word-image embeddings. ECML-2010.

---

Thanks for your response to the comments.
Summary: This paper proposes a graphical model that provides a principled way to learn a similar measure, which can be viewed as a generalization of the common metric learning approach. Experiments show convincing results over the metric learning approaches although scalability could still be an issue for high-dimensional data.

Submitted by Assigned_Reviewer_8

This paper presents a new metric learning method that learns multiple latent components of a similarity measure and the final metric is assumed to be the maximal similarity of all the latent components (each latent component is a separate similarity measure, and if any of them says “similar”, the final similarity measure gives “similar” to a pair of subjects).

Many metric learning methods use real-valued similarity measure which outputs a score that measures the magnitude of the similarity, such as Mahalanobis distance, or probabilistic distance. Hence, even man and centaur, or centaur and horse have relative higher similarity scores, man and horse can have a lower score. However, I agree that it can be important to know and learn there are different latent components (specific ways to compare subjects) among different labelers, and learn what they are that eventually make human annotators label the subjects in such a way. The proposed method is interesting and can be useful.

On page 3, under “latent components”, M_k is sparse, but according to the context, it should use those features at the [k]-th rows and columns, but why those entries are zeros?

The paper omits very important derivation about how to learn \theta on page 4, and focus on the learning of M_k and b_k, but the formula Eq(8) used to calculate them is related to the part that is omitted.

Further, the method changed to use a symmetric decomposition of M_k and stated that they observe no significant detrimental effect of arriving at those solutions. It is better to include some justification and evidence.

If M_k is degenerated into a diagonal matrix, the method is similar to those methods that perform feature selection for metric learning (although the proposed method may result in multiple subsets of features), and should be compared with some of those methods in terms of classification performance on real data.
Summary: This paper presents a new metric learning method that learns multiple latent components of a similarity measure and the final metric is assumed to be the maximal similarity of all the latent components. It can be a useful method.
Author Feedback

Author rebuttal: We thank all the reviewers for their comments. We are especially pleased with their recognition of the proposed method being interesting and clearly motivated as well as the experimental results being convincing.

We clarify several common comments, followed by individual ones.

The example of CENTAUR, MAN and HORSE

We appreciate all the suggestions (including the extra example and references) and will rephrase this part more precisely and clearly. The example is to show that similarity is a broad notion (for example, non-transitivity and non-metricity), thus modeling it with a metric function is limiting.

Sparsity of parameters

We will fix the typo in Line 125-126: “We restrict M_k to be sparse, in particular, the corresponding [k]-th rows and columns are zeroes”. The “zeroes” should be “non-zeroes”.

Further clarifications/discussions on why M_k is sparse naturally when learning: we believe there are two forces at play. Much of the existing work on metric learning has shown that the metrics being learned are often low-rank (without being regularized explicitly). That is leveraged in our work by casting components as metric-based models. Secondly, we use the (noisy-)OR gate to combine local similarity values. Thus, if one component is able to predict similarity, the other components are “suppressed” -- they will receive little learning signals (their similarity values are less relevant to the final outcome). This is evidenced by the component-wise activation patterns in Fig. 3 where we show that each component is often activated specifically.


===R6 ===

Comparison to LMNN in sec 3.1: yes, we threshold the distance computed with the LMNN metric. The threshold is tuned on a development dataset. The synthetic dataset was generated without using labels so the similarity would be considered to be “noisy” when compared to label-induced similarity.

Experiments: we appreciate the suggestions and will incorporate them in future texts.

=== R8 ===

Eq. (8) describes how M_k and b_k are to be estimated, which is decoupled from how \theta_k are to be estimated (Please refer to the supplementary material S1.1). We will make it clear in the text.

Decomposition of M_k: this is often used in existing metric learning work to speed up optimization. The intuition is that in most cases, the local optimum is actually globally optimal (cf. Burer and Monteiro. Math. Programming. 2003)

Feature selection: this is an interesting angle to pursue. Thank you for suggesting that.